# 50 Shades of Deceptive Patterns: A Unified Taxonomy, Multimodal Detection, and Security Implications

## ABSTRACT

Deceptive patterns (DPs) are user interface designs deliberately crafted to manipulate users into unintended decisions, often by exploiting cognitive biases for the benefit of companies or services. While numerous studies have explored ways to identify these deceptive patterns, many existing solutions require significant human intervention and struggle to keep pace with the evolving nature of deceptive designs. To address these challenges, we expanded the deceptive pattern taxonomy from security and privacy perspectives, refining its categories and scope. We created a comprehensive dataset of deceptive patterns by integrating existing small-scale datasets with new samples, resulting in 6,725 images and 10,421 DP instances from mobile apps and websites. We then developed DP-Guard, a novel automatic tool leveraging commercial multimodal large language models (MLLMs) for deceptive pattern detection. Experimental results show that DPGuard outperforms state-of-the-art methods. Finally, we conducted an extensive empirical evaluation on 2,000 popular mobile apps and websites, revealing that 23.61% of mobile screenshots and 47.27% of website screenshots feature at least one deceptive pattern instance. Through four unexplored case studies that inform security implications, we highlight the critical importance of the unified taxonomy in addressing the growing challenges of Internet deception.

## 1 INTRODUCTION

Mobile apps and websites are pervasive in people's daily lives, through which people can accomplish different tasks such as chatting, learning, gaming, browsing news, and shopping. However, many forms of deception are embedded within these services. While people are often aware of and take action against scams, there are more subtle deceptions integrated into the user interfaces they interact with daily [11, 27]. These types of deception are referred to as deceptive patterns, also known as dark patterns. They exploit cognitive biases through visual and linguistic manipulation, tricking users into taking actions that may undermine their interests [8, 11, 15, 26]. For example, the `Forced Continuity` pattern offers a free trial for app subscriptions while concealing the fact that the subscription will automatically renew. The `Privacy Zuckering` pattern buries related terms of use in lengthy documents or makes them difficult to find, often using with `Preselection` patterns to mislead users and collect personal information without their awareness. Falling for these tricks can not only lead to psychological stress [5, 31], financial loss [46], and privacy breaches [3, 29, 42] but also diminish user autonomy, ultimately compromising their well-being and control over digital interactions and eroding digital trust [5].

Efforts have been made to better understand deceptive patterns through empirical studies, with the goal of establishing a comprehensive taxonomy and reporting on current practices [7, 11, 15, 21, 27, 37]. A plethora of research has enriched the taxonomy across different platforms [11, 21, 27] and languages [18]. Recognizing inconsistencies across these taxonomies, recent works [8, 26, 30] sought to integrate them into a unified framework, facilitating consistent understanding and enabling automated detection. However, these efforts primarily focus on the general characteristics of deceptive patterns, often overlooking the privacy and security implications, which have the potential to cause far more severe consequences. Moreover, these studies are often limited in scale [11, 15], which may not fully capture the broader distribution of deceptive patterns, or are restricted to specific domains [27].

Meanwhile, legal entities are actively working to establish frameworks for regulating and limiting the use of deceptive patterns [4, 12, 32]. However, with the sheer volume of apps and websites available – and the constant emergence of new ones and daily updates – it is neither feasible nor practical for regulators to examine them all. Fortunately, some attempt to automate the detection process using machine learning methods [8, 26]. Mansur *et al.* [26] and Chen *et al.* [8] concurrently proposed the first automated methods to leverage machine learning methods to first extract meta data from a single user interface (UI) screenshot, and then use rule-based methods to identify the existence of deceptive patterns. However, their rule-based methods have several limitations. First, as the taxonomy evolves over time, these methods require significant maintenance efforts to adapt to new changes. Second, even with manual updates, rigid rules that lack an understanding of UI semantics are not robust enough, often failing to capture the complexity and subtlety of deceptive patterns. For example, a `Hidden Cost` pattern (as shown in Figure 9(b) in Appendix L) might offer a free service for 7 days while concealing the price information that applies after the trial period ends. Existing rule-based approaches struggle to detect such deceptive patterns because the hidden information is not explicitly shown. A nuanced change of text patterns will greatly deteriorate the performance of their methods.

To bridge the gaps in current research, we begin by systematically analyzing existing taxonomies and develop a unified framework, incorporating both category and scope-level considerations (Section 3). Building on this, we create a comprehensive, cross-platform dataset by merging existing datasets and incorporating the latest trends in deceptive patterns. As a result, we collected 5,059 UIs from existing sources and manually added 1,666 trendy UIs, resulting in a rich dataset of 3,348 deceptive UI images with 7,044 deceptive pattern instances, alongside 3,377 non-deceptive UI images (*i.e.,* benign images) *(Note: one DP image may contain multiple DP instances or patterns)*.

To support robust detection, we propose a deceptive pattern guard, DPGuard, a novel framework that combines the strengths of a mature classification model with advanced multimodal large language models (MLLM) to capture subtle nuances in UIs and perform effective detection. DPGuard significantly reduces the burden of manual inspection by enabling the model to automatically learn

**Table 1: A summary of recent studies on deceptive patterns.**

| Studies | Platforms | | Key Contributions | | | Year | # of DP Instances |
|---|---|---|---|---|---|---|---|
| | Web | Mobile | Taxonomy | Dataset | Detector | | |
| Brignull et al. [7] | ● | ○ | ● | ● | ○ | 2010 | 57 |
| Gray et al. [15] | ○ | ● | ● | ● | ○ | 2018 | 112 |
| Mathur et al. [27] | ● | ○ | ● | ● | ● | 2019 | 1,818 |
| LLE [11] | ○ | ● | ● | ● | ○ | 2020 | 1,787 |
| Nazarov et al. [28] | ● | ○ | ○ | ○ | ● | 2022 | - |
| AidUI [26] | ● | ● | ● | ● | ● | 2023 | 301 |
| UIGuard [8] | ○ | ● | ● | ● | ● | 2023 | 1,600 |
| Nie et al. [30] | ● | ● | ● | ○ | ○ | 2024 | - |
| **DPGuard (ours)** | ● | ● | ● | ● | ● | 2024 | 7,044 |

● (○): the item is (not) supported by the study; -: not applicable.

and identify intricate patterns. Specifically, given a UI, it is first processed by a binary classifier to determine whether deceptive patterns are present. If classified as deceptive, the MLLM is employed to identify the specific type of a deceptive pattern. In addition, we introduce a novel prompting mutation technique, allowing the MLLM to iteratively refine its prompts and accurately identify the key features associated with each deceptive pattern. This adaptive prompting method ensures that DPGuard can efficiently handle diverse scenarios, offering a robust and scalable solution for detecting deceptive patterns across different platforms. To evaluate the effectiveness of our proposed system, we evaluate each component and the overall performance of DPGuard on the newly created dataset. We further conducted an empirical evaluation on 2,000 UI images collected from popular mobile apps and websites. In summary, our contributions are as follows:

- We develop a unified deceptive pattern taxonomy by systematically analyzing existing ones, incorporating privacy and security aspects from both category and scope-level refinements.
- We contribute a comprehensive, cross-platform deceptive pattern dataset that captures both the most up-to-date data and a multi-year timeline, offering rich insights for analyzing trends and patterns over time.
- We propose DPGuard, a hybrid approach that combines a binary classifier with a multimodal language model, which leverages a prompt mutation strategy to optimize the optimal prompt, achieving state-of-the-art performance.
- We conduct extensive experiments to evaluate the performance of DPGuard, and provide unexplored case studies that inform security implications.

## 2 RELATED WORK

In this section, we introduce recent studies on deceptive patterns, including works related to DP taxonomies and detection methods. We summarized these studies in Table 1.

**Deceptive pattern taxonomies.** A pioneering work in deceptive pattern research is a wiki-like website launched in 2010 [7], where 14 deceptive pattern categories are defined with example showcases, which also encourage end-users to report deceptive patterns encountered in daily life via Twitter [6]. Mathur et al. [27] proposed a refined taxonomy with 7 core categories and 15 subcategories, analyzing 11K shopping websites. Later, AidUI, one of the state-of-the-art (SOTA) deceptive pattern detection models, merged the taxonomies [7, 15, 27] into a new taxonomy with 7 core categories and 27 subcategories. Concurrently, Chen et al. [8] integrated existing deceptive pattern taxonomies, creating a unified taxonomy

with 5 core categories and 19 subcategories. These studies highlight the pressing need for the taxonomy of deceptive patterns to evolve as new UI designs and deceptive practices emerge with advancing technologies. A "concept drift" issue may also arise, as changes in deceptive practices can render existing taxonomies outdated. Furthermore, most current research focuses primarily on deceptive patterns from a UI design perspective, overlooking the significant security and privacy risks these patterns pose. To address this gap, we refine the taxonomy to not only account for evolving deceptive practices but also incorporate categories specifically related to security and privacy, offering a more comprehensive classification suited to contemporary contexts.

**Deceptive pattern detection.** To protect end-users' best interests and mitigate the risk of deception, various clustering algorithms have been initially proposed to group similar deceptive pattern (DP) images, which are then manually labeled as DP categories. UIGuard [8] and AidUI [26] are two examples of approaches that combining both deep learning and manual-defined rules for detection. Specifically, UIGuard extracts property features, such as element types and coordinates, and element relationships, from Android app screenshots, through several deep learning models, and then uses a rule table as a knowledge base to identify areas where deceptive patterns may exist. Similarly, AidUI first detects visual cues and extracts text information from UI images, then analyzes deceptive patterns based on spatial, textual, and color features. AidUI further performs both segment-level and UI-level resolution to localize deceptive patterns. These approaches, while valuable, currently require significant manual effort for cluster labeling and maintaining rule-based knowledge systems, which can make it challenging to adapt to concept drift. To streamline this process and more effectively address evolving deceptive patterns, we propose a new detection method that utilizes a multimodal large language model. This multimodal approach allows for seamless adaptation to new patterns with minimal effort, significantly reducing the need for extensive human intervention.

## 3 TAXONOMY REFINEMENT

Deceptive patterns were initially viewed as a human-computer interaction problem, defined as maliciously designed interfaces that mislead users into unintended actions [11]. Consequently, previous taxonomies often overlook security- and privacy-related examples. However, since some deceptive UI designs can lead to serious security or privacy consequences, such as forced enrollment do not provide "back" or "cancel" button so user can not make their own decision but to submit their personal information to use the provided service, but have not yet been included in a specific DP category. We extended existing taxonomies [8] in this study to minimize such omissions. We reviewed and summarized taxonomies from recent studies [8, 11, 26], examining the definitions of DP categories and identifying overlaps or contradictions. To develop a more fine-grained taxonomy that can be effectively applied to DP detection, we refined the UIGuard taxonomy, as it is a pioneering and SOTA work that integrates existing taxonomies into a unified framework. Specifically, we refine the DP taxonomy at two levels: category-level (adding or removing DP categories) and scope-level (updating the scope of existing DP categories). We present our DP

Table 2: Statistics for the new deceptive pattern dataset.

| Categories | Definitions | Cases | # of Samples Collected | | |
|---|---|---|---|---|---|
| | | | Mobile | Website | Subtotal |
| 0 - No DP | No deceptive pattern | - | 3,018 | 359 | 3,377 |
| 1 - Nagging | An unexpected pop-up window keeps appearing repeatedly, disrupting the user's activities. | Pop-up Ads; pop-up to rate; pop-up to upgrade | 410 | 183 | 593 |
| 2 - Roach Motel | Easy to opt-in, but impossible or hard to opt out. | Unable/hard to unsubscribe some services; unable/hard to get refund; unable/hard to delete account | 24 | 13 | 37 |
| 3 - Price Comparison Prevention | Making a direct comparison with others is difficult | Unable to copy and paste product name; unable to compare all other plans at the same time | 7 | 27 | 34 |
| 4 - Intermediate Currency | Users are distanced from real money by being prompted to buy virtual currencies | Purchasing virtual coins, diamonds, gems, or credits is required to continue using certain internal services. | 38 | 6 | 44 |
| 5 - Forced Continuity | Users are still charged after the service has expired. | The subscription will automatically renew after the free trial or discount period ends | 51 | 29 | 80 |
| 6 - Hidden Costs | The costs are not disclosed at the initial stage | Delivery fee, shipping fee, service fee, tax fee, subscription fee are not shown initially | 38 | 113 | 151 |
| 7 - Sneak into Basket | Additional charged items are added without the user's selection | A donation will be added to the bill to round it up; a charged service, such as insurance, will be added to the bill | 1 | 6 | 7 |
| 8 - Hidden Information | Options or actions are made difficult for the user to read or understand immediately | Relevant information, such as terms of service, is displayed in small, greyed-out text; use hyperlinks for relevant information (e.g., terms of service, agreement) | 242 | 399 | 641 |
| 9 - Preselection | Some choices are preselected by default | Unnecessary options are preselected (e.g., cookies, data sharing, policy, terms, agreement, notification); the expensive plan is preselected by default | 367 | 430 | 797 |
| 10 - Toying with Emotion | Language, color, and style are used to evoke emotions, pressuring users into taking a certain action | Countdown timer/limited rewards; confirm shaming; fake scarcity (high demand, low stock) | 86 | 251 | 337 |
| 11 - False Hierarchy | One option is made more prominent than other equally available options | One button is more salient than the other (e.g., accept and close button) | 561 | 341 | 902 |
| 12 - Disguised Ad | Ads pretend to be normal content | Sponsored ads or content are disguised as banners or inserted in the normal content | 905 | 382 | 1,287 |
| 13 - Tricked Questions | Confusing or overly complex wording is used to explain something or ask questions | Double negation | 5 | 5 | 10 |
| 14 - Small Close Button | The button to close the current content is hard to identify | The real close ads button is very small or hard to recognize | 752 | 205 | 957 |
| 15 - Social Pyramid | Users are prompted to share something with friends to receive rewards or unlock features | Share unnecessary information with friends; invite friends to get vouchers/credits/points/prizes | 36 | 7 | 43 |
| 16 - Privacy Zuckering | Unnecessary information is collected by default | Forced to agree to agreements (e.g., terms of use or privacy policy) before using the service | 210 | 374 | 584 |
| 17 - Gamification | Requires users to repeatedly perform actions to get something | Daily check-in rewards, lucky wheel | 27 | 1 | 28 |
| 18 - Countdown on Ads | Ads can only be closed once the countdown timer reaches zero | The countdown timer on the ads | 77 | 10 | 87 |
| 19 - Watch Ads to Unlock Features or Rewards | Unlock features or get rewards by watching ads | Users are required to watch ads to access or unlock a tool, service, or feature | 71 | 0 | 71 |
| 20 - Pay to Avoid Ads | Using money to remove ads | Upgrade to the pro version or subscribe to a paid plan to remove ads; pay for a service to eliminate ads | 108 | 7 | 115 |
| 21 - Forced Enrollment | Users are required to sign up or sign in before they can access the service | Users are required to sign up or sign in on the application's home page before they can perform further actions; users are required to sign up or sign in before they can continue viewing the content | 150 | 89 | 239 |
| **Total Instances** | | | **7,184** | **3,237** | **10,421** |

taxonomy in Table 2, with specific refinements highlighted in blue. This taxonomy includes 21 DP categories across 33 use cases, designed to capture a wide range of security- and privacy-related examples.

**Category-level refinement.** In the category-level refinement, We made two modifications based on the 19 deceptive pattern categories introduced in UIGuard.

We removed Bait-and-Switch category. According to the definition by Brignull *et al.* [7], Bait-and-Switch refers to a situation where the user performs an action but receives an undesired result. Although many studies [7, 8, 11, 15, 26, 28, 30] include this category in their taxonomy, only two of them [7, 15] collectively reported 10 instances of this type, and another work [30] instead considered the Disguised ads as a case of Bait and Switch. We believe the rarity of Bait-and-Switch examples is due to its overly broad definition of an "undesired result", which is often addressed by other deceptive pattern categories with more specific definitions.

For example, as shown in Figure 9(a) Appendix L, if a user clicks a UI element but is shown an ad instead, the case could be classified as Disguised Ads. If the user repeatedly clicks on UI elements and always receives ads, it would typically be reported as Nagging. Additionally, we found that the concept of Bait-and-Switch overlaps with other categories like Small Close Button, Watch Ads to Unlock Features or Rewards, Hidden Costs, and Hidden Information. Therefore, we removed the Bait-and-Switch category to simplify the taxonomy. We reviewed the 10 instances from prior works [7, 15], and identified all of them fall within these scenarios.

We reintroduced Forced Enrollment into the taxonomy. In previous studies, Forced Enrollment refers to situations where users are required to sign up or sign in to use a service, even when enrollment is unnecessary. Such DP can lead to serious security or privacy concerns, as it often forces users to provide additional personal information to access the service. Unlike prior work, our

taxonomy specifically includes a constraint: we only classify it as "forced" enrollment when the user cannot skip the sign-up or sign-in process, *e.g.,* there are no "Go back" or "Cancel" buttons available in the UI.

*Seriousness analysis on* `Forced Enrollment`*:* In the Alice (end-user) and Bob (service provider) model, Bob poses a potential privacy risk if he mishandles or over-collects Alice's data. If Alice is forced to disclose sensitive information, her security goals—such as confidentiality and privacy—may be compromised. Eve (an eavesdropper) could exploit this data to perform attacks like identity theft, phishing, or unauthorized surveillance. Mallory (a malicious actor) could further exploit the stored personal data to impersonate Alice, commit financial fraud, or launch other types of attacks. In some cases, Trent (a trusted third party) might be involved in managing data or providing identity verification. However, if Alice is forced to enroll, she may not trust Trent or Bob to securely handle her data or ensure its use for legitimate purposes. This lack of trust introduces vulnerabilities where Alice's data could be mishandled or misused by Bob, Trent, or even Eve and Mallory if their systems are compromised.

**Scope-level refinement.** In the scope-level refinement, we retain the existing deceptive pattern categories but expand some of their scope to include additional security- and privacy-related examples. Specifically, we expand the scope of five categories, *i.e.,* `Price Comparison Prevention`, `Hidden Cost`, `Hidden Information`, `Toying with Emotion`, and `Disguised Ads`). We showcase one example of `Disguised Ads` here and discuss the rest in Section 6 for in-depth analysis.

Chen *et al.* [8] defined `Disguised Ads` as DP instances where developers present the sponsored ad pretending to be a normal content and place it in the middle of the screen. However, we found that the top and bottom ads banner should also be included in this category as these instances are also disguised as normal content, along with the potential risks they post, regardless of their placement. See Figure 9(a) in Appendix L for an example.

*Seriousness Analysis on* `Disguised Ads`*:* In the Alice and Bob model, if Alice accidentally clicks on a disguised advertisement (ad), she could be unknowingly redirected to a website or service controlled by Eve. By tricking Alice into clicking the ad, Eve (the eavesdropper and advertiser) can collect Alice's data, such as her device information, browsing habits, or personal identifiers. Eve may then track Alice across different websites or apps, building a user profile and potentially violating her privacy. If the disguised ad contains malicious content, Mallory could exploit Alice's accidental click to carry out malicious actions, such as phishing attacks or malware installation. In this scenario, Trent, the app platform (*e.g.,* Google Play), is expected to enforce clear labeling of ads to prevent user deception. However, if Trent fails to strictly enforce these guidelines, apps like Bob's could continue using deceptive ads, eroding user trust and compromising security.

**Dataset creation.** Overall, the unified taxonomy is underpinned by the new dataset collated in principle through (*i*) merging and annotating existing datasets and (*ii*) incorporating new, up-to-date DP examples from additional resources (see Appendix for concrete steps for data curation). The unified taxonomy ultimately includes 10,421 deceptive pattern instances, addressing the issue of scale. To ensure comprehensiveness, our dataset features 5,269 images

**Table 3: Statistics of dataset collection.**

| Data Sources | | # of Instances (UI Images) | | | |
|---|---|---|---|---|---|
| | | Mobile | | Website | |
| | | DP | non-DP | DP | non-DP |
| From Existing Datasets | UIGuard | 2,253 (1,204) | 2,757 (2,757) | 0 (0) | 0 (0) |
| | AidUI | 332 (208) | 110 (110) | 167 (94) | 59 (59) |
| | LLE | 865 (477) | 150 (150) | 0 (0) | 0 (0) |
| New Samples | WebUI | 0 (0) | 0 (0) | 1,806 (643) | 299 (299) |
| | Popular Lists | 716 (362) | 1 (1) | 905 (360) | 1 (1) |
| **Total** | | 4,166 (2,251) | 3,018 (3,018) | 2,878 (1,097) | 359 (359) |

from mobile platforms, containing 7,184 instances, and 1,456 images from websites, containing 3,237 deceptive pattern instances. The dataset spans images from 2017 to 2024, ensure it is up-to-date. The breakdown statistics of the sources of the dataset are reported in Table 3. Additionally, our dataset also ensures every category has at least 5 representative examples, which reduces the learning curve for future research.

## 4 DPGUARD: DETECTION OF DECEPTIVE PATTERNS

In this section, we propose a novel framework for DP detection, which involves a hybrid approach that combines a binary classifier with an MLLM, aiming to achieve SOTA performance.

### 4.1 Overview

To reduce manual efforts in deceptive pattern detection and address the challenge of concept drift, we propose an automatic framework that utilizes an MLLM for deceptive pattern detection. Specifically, we design a mutation-based prompt engineering approach to enhance the MLLM's performance in the task of DP detection. Additionally, to lower the practical cost of DP detection, we incorporate a binary classifier *before* the MLLM module to determine whether DP is present in the samples to be examined, ensuring that only highly suspicious ones are passed to the MLLM for further analysis. Figure 1 presents an overview of the proposed DPGuard framework, detailing the process during offline training stages (binary classifier training, prompt mutation with MLMM), and online inference stages.

### 4.2 Binary Classifier

To ensure practicality and reduce costs, we first introduce a binary classifier to filter out images with a low probability of being deceptive. To achieve a high-performance binary classifier, we train or fine-tune several SOTA machine learning models or open-source large language models, selecting the best-performing one based on its F1-score on the test dataset. Further details of the models and experiments can be seen in Section 5.1. As a result, we used the ResNet101 [17] model, replacing the last output layer as a binary projection layer, and fine-tuning all layers.

During the inference phase, if the binary classifier determines that the provided image does not contain any deceptive patterns, we classify it as a non-DP image. Otherwise, the image is sent to the MLLM for further categorical examination.

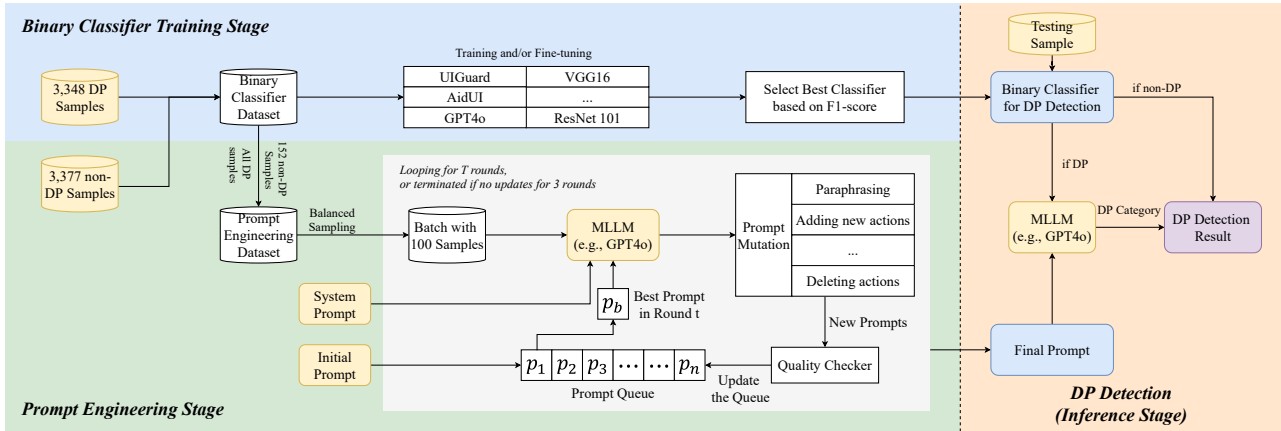

**Figure 1: An overview of DPGuard framework.**

---

**Algorithm 1:** A Mutation-based Prompt Engineering

**Input:** Initial prompt $P_0$, System prompt $P_s$, Mutation instructions $P_m$, Prompt queue $Q$, Queue size limit $n$, Number of new prompts to be generated in each round $m$, Training dataset $D$, Total mutation rounds $T$, Multimodal language model $M$, Batch size $b$, Similarity threshold $s$.

**Output:** Best prompt $p_b$.

1   $p_b \leftarrow P_0, Q \leftarrow \emptyset, t \leftarrow 0$;
2   $Q$.enqueue($P_0$);
3   **while** $t < T$ **do**
4      $q \leftarrow \emptyset, l \leftarrow n + m - Q$.size;
5      **while** $q.length < l$ **do**
6         $p \leftarrow M(P_m, P_s, p_b)$;
7         **if** $similarity(p, P_0) > s$ **then**
8            $q$.attend($p$);
9      $Q$.extend($q$);
10     $(X, Y) \leftarrow$ random_sampling($D, b$);
11     $Y' \leftarrow M(P_s, Q, X)$;
12     $Q$.sort(Loss($Y, Y'$));
13     $Q \leftarrow Q[1 : n]$;
14     $p_b \leftarrow Q[0]$;
15     $t \leftarrow t + 1$;
16   **return** $p_b$

---

## 4.3 Prompt Engineering

For the selection of the MLLM, we choose a commercial MLLM as the baseline, as recent research reports that commercial MLLMs, such as GPT4 [34], Gemini [10], and Claude [2], outperform open-source ones [9, 25, 48] in several downstream tasks [23]. However, these commercial MLLMs do not support fine-tuning with the image modality, so we proposed a prompt engineering strategy to boost MLLM's performance in the specific task of DP detection, which involves three key steps: prompt mutation, prompt quality checker, and the maintenance of a prompt queue. We illustrate our prompt engineering method in Algorithm 1.

**Prompt mutation.** Prompt mutation refers to a technique in prompt engineering where variations or modifications are systematically applied to an original prompt to enhance the performance of large language models in specific tasks. In this study, we revise PromptBreeder [13] and adapt it to the specific nature of DP detection tasks. Specifically, we incorporate domain-specific knowledge into the revised prompting strategy, enabling its application on large datasets. Additionally, our prompt mutation approach leverages GPT's randomness in prompt generation and is designed to work effectively across multiple modalities, including both text and image. We first manually define the system prompt $P_s$ (which provides some domain knowledge, such as descriptions for all DP categories according to the definition in Table 2) and a simple initial prompt $P_0$ (see Appendix E) asking MLLM to "detect if any deceptive pattern in an image". In each mutation round, we ask MLLM to generate new prompts by performing one of the following actions defined in mutation instructions $P_m$: paraphrasing the current prompt, adding some new actions (*e.g.,* check UI color, check UI text) or delete actions that may mislead detection (Line 6 in Algorithm 1). Before the generated prompts are accepted, mutation criteria are applied to ensure the prompts are of high quality.

**Mutation criteria.** Since prompt mutation in our architecture is driven by randomness, the mutation criteria are designed to ensure that prompts evolve in the desired direction, *e.g.,* preserving key semantics. Specifically, we use a sentence transformer [36] to encode each prompt and calculate the cosine similarity between the generated prompts at each mutation round and the initial prompt, $P_0$ (Line 7 in Algorithm 1). To determine the similarity threshold $s$, we conducted a pilot study by generating 90 prompts (30 mutated prompts over 3 rounds), manually grouping them into good- and poor-quality categories, and calculating the cosine similarities to $P_0$ across the groups. We then identified a threshold that best separates the two groups, which is set as 0.2.

**Maintain the prompt queue.** Considering that the cost of querying MLLM in a multimodal manner is significantly higher than for text-only queries, we randomly sampled 100 examples in a balanced manner (ensuring approximately equal samples from each DP category), rather than using the entire dataset, in each mutation round to evaluate the performance of the newly generated prompts

(Line 10 in Algorithm 1). However, this strategy may not yield precise performance results. Therefore, we decided to maintain a prompt queue as a buffer to select the best prompts across multiple rounds of mutations.

As shown in Lines 11 to 13 in Algorithm 1, we sort the prompts in the queue (including prompts stored in the last mutation rounds and newly generated prompts) according to their loss against the ground truth. The loss function we use is Binary Cross Entropy Loss [35]. After sorting, we retain the top $n$ prompts in the queue and select the first prompt as the best one, $p_b$. In the next mutation round, new prompts will be generated based on $p_b$, continuing this process until the maximum number of mutation rounds, $T$, is reached. Note that if $p_b$ has not been updated for 3 iterations, we assume that the optimal prompt has been found and will terminate the training process. After rounds of mutation and selection, the best prompt $p_b$ will be recorded and used in the inference phase to infer the type of deceptive patterns.

## 5 EVALUATION

In this section, we evaluate the performance of DPGuard through comprehensive experiments and conduct an empirical study to assess its effectiveness in real-world scenarios. The details on building the new DP dataset and the process for collecting samples for the empirical study are provided in Appendix D.2.

### 5.1 DPGuard Performance

To evaluate the effectiveness of DPGuard in DP detection, we first report the module-level performance of the binary classifier and MLLM on DP and non-DP instances respectively, and then present the overall performance of the entire framework.

**DP detection through a binary classifier.** According to Section 4.2, we analysed the performance of different binary classifiers, including inference-only models, such as commercial MLLMs (OpenAI GPT Series [33, 34]), the SOTA model UIGuard [8] and AidUI [26], and trainable models (*e.g.,* VGG [39], DenseNet [19], and ResNet [17]). Specifically, for trainable/finetunable models, we first perform data pre-processing (*e.g.,* resizing the image and applying image embedding normalization), and then obtain the final layer's embedding from the model. We then mapped its shape to 2 and applied the sigmoid function to make a binary prediction. The dataset used for fine-tuning the binary classifier is described in Appendix C. We use the training set for fine-tuning and updating hyper-parameters, the validation set for selecting the best model, and the testing set for evaluating the final performance for fine-tuned model. Table 4 reports the binary classification results, with the pre-trained ResNet101 model achieving the best F1 performance (exceeding 0.87). We believe this represents satisfactory performance, surpassing existing DP detection tools such as UIGuard and AidUI. However, some DP examples may still be misclassified as non-DP, which we view as a reasonable trade-off between practical costs and detection performance.

> **Takeaway 1:** *A satisfactory binary classification on DP can be achieved by fine-tuning pre-trained CNN models. However, incorporating more accurate models into the framework could further enhance overall performance.*

**Table 4: Performance of candidate binary classifiers.**

| Model | On DP Instances | | | On non-DP Instances | | |
|---|---|---|---|---|---|---|
| | Precision | Recall | F1 | Precision | Recall | F1 |
| GPT4o | 0.3805 | 0.7997 | 0.5156 | 0.9516 | 0.7515 | 0.8398 |
| GPT4o-mini | 0.3584 | 0.6041 | 0.4499 | 0.9131 | 0.7936 | 0.8492 |
| UIGuard | 0.3298 | 0.7088 | 0.4501 | 0.8788 | 0.5944 | 0.7091 |
| AidUI | 0.6473 | 0.8392 | 0.7308 | 0.7073 | 0.4595 | 0.5571 |
| VGG16 | 0.6433 | 0.8096 | 0.7169 | 0.7666 | 0.5821 | 0.6618 |
| DenseNet121 | 0.7346 | 0.6811 | 0.7068 | 0.7220 | 0.7709 | 0.7456 |
| ResNet50 | 0.8808 | 0.8467 | 0.8635 | 0.8623 | 0.8934 | 0.8776 |
| ResNet101 | 0.8638 | 0.8839 | **0.8769** | 0.8895 | 0.8703 | **0.8798** |

**Table 5: MLLM performance on DP category determining.**

| Approaches | Metrics | Precision | Recall | F1 |
|---|---|---|---|---|
| Fixed-prompt | Micro avg | 0.4974 | 0.5045 | 0.5009 |
| | Macro avg | 0.3370 | 0.4768 | 0.3522 |
| Prompt mutation | Micro avg | 0.4966 | 0.5577 | **0.5254** |
| | Macro avg | 0.3904 | 0.5054 | **0.4131** |

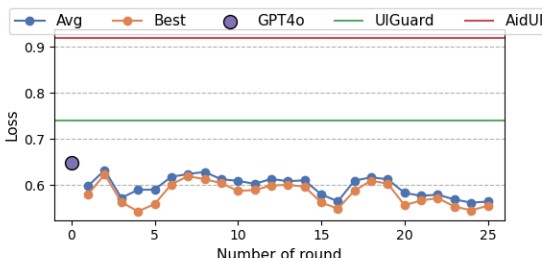

**Figure 2: Loss in each mutation round.**

**DP category determining through MLLM.** In our prompt mutation strategy, there are three hyper-parameters: similarity threshold, queue limits, and the number of mutation rounds. We select these hyper-parameters through several pilot experiments. Please find more details in Appendix J. To demonstrate the performance of our prompt engineering, we use GPT4o with a fixed initial prompt as a baseline, and compare it to the final best prompt after 25 rounds of mutation, under queue size 15. We provide the initial prompt and the final best prompt in Appendix E and Appendix F. As shown in Table 5, the final prompt achieves 2.5% and 6.1% higher F1 scores on both micro and macro averages, respectively. The results are evaluated on 3,490 DP images (6,841 DP instances) from the prompt mutation dataset. We further report the performance of the best prompt in each round in Figure 2 in Appendix J. Due to the randomness present in MLLM outputs and the fact that we evaluate prompts and maintain the prompt queue using a batch of samples (*e.g.,* 100 samples) in each round for cost considerations, we acknowledge that the performance of prompts may vary across mutation rounds, and additional rounds of mutation may yield better prompts. However, the loss of our method (using a prompt mutation strategy to enhance MLLM) for DP detection remains lower than that of the SOTA methods, such as UIGuard.

**Overall performance of DPGuard.** We then compared the overall performance of DPGuard with two existing deceptive pattern detection tools. The key results, including the micro and macro

**Table 6: Performance (F1-score) comparison between DP-Guard, UIGuard, and AidUI on mobile and website datasets.**

| DP Categories | Mobile | | | | Website | | |
|---|---|---|---|---|---|---|---|
| | Instances | UIGuard | AidUI | DPGuard | Instances | AidUI | DPGuard |
| No DP | 3,018 | 0.8091 | 0.7812 | **0.9807** | 359 | 0.4338 | **0.8230** |
| Nagging | 409 | **0.4412** | 0.3454 | 0.3876 | 180 | 0.1163 | **0.4945** |
| Roach Motel | 24 | - | - | **0.5484** | 13 | - | **0.4000** |
| Price Comparison Prevention | 7 | - | - | 0.0000 | 27 | - | **0.2381** |
| Intermediate Currency | 38 | - | - | **0.6154** | 5 | - | **0.4286** |
| Forced Continuity | 48 | 0.0408 | - | **0.7059** | 26 | - | **0.3448** |
| Hidden Costs | 38 | - | - | **0.2680** | 99 | - | **0.2519** |
| Hidden Information | 236 | - | - | **0.4187** | 377 | - | **0.4535** |
| Preselection | 356 | 0.4546 | 0.3565 | **0.5466** | 413 | **0.3629** | 0.2753 |
| Toying with Emotion | 84 | - | 0.1389 | 0.3096 | 229 | 0.4251 | **0.5866** |
| False Hierarchy | 559 | 0.4188 | 0.0552 | **0.6535** | 320 | 0.0245 | **0.4360** |
| Disguised Ad | 883 | 0.1520 | 0.2551 | **0.8481** | 256 | 0.2096 | **0.8060** |
| Small Close Button | 747 | **0.9410** | - | 0.4906 | 160 | - | **0.2564** |
| Social Pyramid | 35 | **0.6349** | - | 0.5047 | 7 | - | **0.3243** |
| Privacy Zuckering | 206 | **0.7378** | - | 0.4073 | 367 | - | **0.5868** |
| Gamification | 27 | 0.3529 | - | **0.5000** | 1 | 0.0000 | 0.0000 |
| Countdown on Ads | 77 | 0.2128 | 0.0000 | **0.3952** | 10 | - | **0.4103** |
| Watch Ads to Unlock Features or Rewards | 67 | **0.3488** | - | 0.0000 | 0 | - | 0.0000 |
| Pay to Avoid Ads | 106 | **0.7265** | - | 0.6277 | 7 | - | **0.1429** |
| Forced Enrollment | 149 | - | - | **0.4383** | 89 | - | **0.3356** |
| Micro avg | 7,114 | 0.6672 | 0.5889 | **0.7316** | 2,945 | 0.3228 | **0.4989** |
| Macro avg | 7,114 | 0.2851 | 0.0878 | **0.4385** | 2,945 | 0.0715 | **0.3452** |

-: the DP category is not supported by the corresponding tool.

averages of F1-scores, are detailed in Table 6. More detailed results with precision and recall are provided in Tables 11 and 12 in Appendix K. Specifically, DPGuard achieves 0.6326, 0.6927 and 0.6613 on micro averaged precision, recall and F1-score, and 0.4122, 0.5698 and 0.4437 on macro averaged precision, recall and F1-score. The result demonstrates that our DPGuard outperforms two SOTA models across all evaluation metrics, advancing the performance of deceptive pattern detection to a new level.

> **Takeaway 2:** *DPGuard outperforms the state-of-the-art models in DP detection, increasing the F1-score to 0.73 (micro) and 0.44 (macro) on the mobile dataset, and 0.50 (micro) and 0.34 (macro) on the website dataset.*

## 5.2 Empirical Evaluation in the Wild

To offer valuable insights into how often people encounter the potential threat of deceptive patterns, we conducted an empirical study with real-world cases collected from popular mobile apps and websites. We collected 2,905 mobile images and 9,396 website images from 1,000 mobile apps (from AndroidZoo [1]) and 1,000 popular websites (based on Majestic Million list [20]). Details of data collection are provided in Appendix D.2.

An analysis of images from mobile and website platforms reveals that 23.61% of mobile images (686 out of 2,905) contain deceptive patterns, with an average of 1.95 instances per image and standard deviation is 0.88. Among these deceptive mobile UIs, the majority (53.01%) contain two deceptive pattern instances, and 16.62% contain more than two instances. In contrast, 47.27% of website images (4,429 out of 9,369) feature deceptive patterns, with a higher average of 2.87 instances per image and the standard deviation is 1.31. Within these deceptive website UIs, the majority (43.70%) also contain two deceptive pattern instances, but a larger proportion (45.94%) contain more than two instances. Based on these results, we find that websites, on average, employ 23.66% more deceptive pattern instances than mobile platforms. Upon a closer inspection

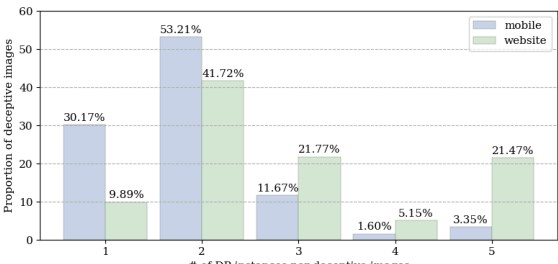

**Figure 3: Distribution of number of deceptive instances per deceptive images.**

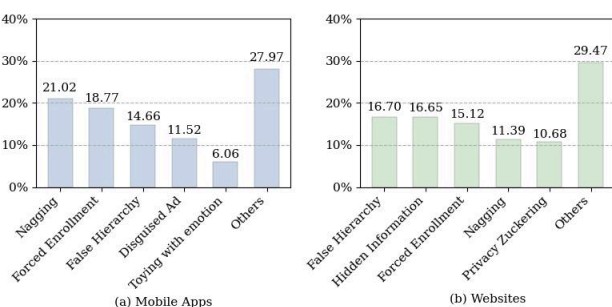

(a) Mobile Apps  (b) Websites

**Figure 4: Distribution of detected deceptive instances in the empirical study.**

on the top categories of DP in mobile apps and websites (see Figure 4 (a) and Figure 4 (b)), the top three most frequent deceptive pattern categories are `False Hierarchy`, `Hidden Information` and `Forced Enrollment` for websites; while for mobile apps, the top 3 DP categories are `Nagging`, `Forced Enrollment` and `False Hierarchy`. We infer that such difference arises because mobile application are more likely to have pop-up ads, while website are more likely to display cookie consent notifications.

To validate the effectiveness of DPGuard in the wild, we randomly selected 20 images each from mobile and website categories, and manually annotated them. As a result, we identified 14 DP images (5 mobile and 9 web) and 26 non-DP images (15 mobile and 11 web). Our binary classifier successfully identified 12 out of 14 (85.71%) deceptive images. Among the identified 12 deceptive images, DPGuard predicted 65 deceptive pattern instances, of which 45 were correct. The performance aligns with our experiments in Section 5.1.

> **Takeaway 3:** *Based on our empirical evaluation, we found that deceptive patterns are frequently present in popular applications. Specifically, 23.61% of mobile app images and 47.27% of website images were identified to contain DPs.*

## 6 CASE STUDY

In this section, we provide unexplored case studies that inform security implications for the expanded deceptive pattern categories.

## 6.1 Price Comparison Prevention

In UIGuard, `Price Comparison Prevention` focuses solely on whether the product name can be copied and pasted for direct

comparison with other markets. However, we found an example (as shown in Figure 9(f) in Appendix L) where a user needs to manually click 'budget server' or 'performance server' to switch between other plans. In the previous deceptive pattern definition, this case is classified as a non-deceptive example, but we believe there are two risks for end-users.

In the context of Alice and Bob's model, Alice is the customer, while Bob represents the service provider. The updated definition of Price comparison prevention now includes any action by Bob that makes it difficult or impossible for Alice to compare different plans or products directly. This deceptive pattern increases the chances of Alice making a purchase decision without fully understanding her options, which could lead to overspending or choosing a plan that does not serve her best interest. For the security risk, Bob's intentional obfuscation might push Alice into sharing her payment information or making a financial commitment under false pretenses. This increased exposure could make Alice vulnerable to financial fraud. For the privacy risk, Bob prevents Alice from comparing prices might force her to engage more with Bob's system, potentially collecting more of her personal data without her realizing the extent of the tracking.

Thus, we believe it is important to expand the definition of price comparison prevention to account for both potential security and privacy risks.

## 6.2 Hidden Cost

The traditional definition of Hidden Cost in deceptive patterns mainly focuses on the additional fee (*e.g.,* deliver fee or service fee) at the checkout stage. However, in people's daily lives, there is a case like the one in Figure 9(b) in Appendix L where the app does not disclose its subscription fee after the free trial. Therefore, we have expanded the scope to include any costs not disclosed at the initial stage, which will now be recognized as a hidden cost deceptive pattern.

In the Alice and Bob model, by not disclosing the subscription fee at the free trial stage, Bob misleads Alice into thinking the service might be free or cheaper than it actually is. This lack of transparency not only risks Alice incurring unexpected financial charges but also increases the likelihood that she will agree to "terms and conditions" without fully understanding them. From a security perspective, this could result in Alice unknowingly providing her payment information, which might be exploited for unauthorized charges later. From a privacy standpoint, since Alice might agree to terms that allow Bob to collect and potentially sell her sensitive information, it creates a significant risk of privacy invasion.

Therefore, the updated definition of hidden costs is a critical deceptive pattern category, as it exposes Alice to financial exploitation and unauthorized data sharing, making her more vulnerable from both security and privacy perspectives.

## 6.3 Hidden Information

The definition of Hidden Information (*i.e.,* a deceptive pattern that makes the options/actions not immediately readable for the user) as described by UIGuard is sufficient, but the use cases need to be expanded to cover the case shown in Figure 9(e) in Appendix L,

where the developer can intentionally hide the privacy-related information in a hyperlink.

By placing privacy-related information within a hyperlink, Bob makes it less likely for Alice to notice or access important details about data usage, risks, or conditions, even though the information is technically provided. This tactic deceives Alice into thinking that there are no significant risks, or at the very least, makes it inconvenient for her to find out. In terms of privacy, Bob could conceal permissions to access, share, or sell Alice's sensitive information in the linked text, thereby tricking her into relinquishing control over her personal data.

Therefore, expanding the use cases for the hidden information category is crucial, as it highlights how such tactics increase the risk of unauthorized data exploitation, making users like Alice vulnerable to privacy breaches.

## 6.4 Toying with Emotion

Toying with emotion in UIGuard is described as using language or visual information to deceptive users into taking action based on emotion. The use cases covered by UIGuard include countdown offer or limited-time rewards. However, we have expanded the use cases for covering some cases such as the one in Fiugre 9(c) in Appendix L, where a developer uses fake scarcity (*e.g.,* high-demand or low-stock) for deceptive user into making irrational decisions.

In Alice and Bob's model, by creating a false sense of scarcity, such as displaying "high-demand" or "low stock" notifications, Bob manipulates Alice's emotions, pushing her to make hurried and irrational decisions without fully considering the consequences. From a security perspective, this sense of urgency may lead Alice to overlook important details, such as verifying the legitimacy of the service or transaction, making her more susceptible to scams or fraudulent activities. In terms of privacy, Alice might quickly provide her personal and financial information without carefully reviewing the terms, potentially exposing her to data misuse or unauthorized sharing.

Therefore, expanding the use cases of the "toying with emotion" deceptive pattern is essential, as it underscores how emotionally charged tactics can lead to impulsive actions, resulting in both financial loss and increased vulnerability to security and privacy breaches for users like Alice.

## 7 CONCLUSION

We have unified the deceptive pattern taxonomy, refining it with 24 subcategories that reflect both category and scope. This updated taxonomy informs the unexplored deception in the wild. We also introduced a novel approach that combines a binary classifier with mutation-based prompt engineering to harness the capabilities of multimodal large language models for deceptive pattern detection. Our experiments demonstrate that DPGuard achieves state-of-the-art performance in this area. We also provided unexplored real case studies with security implications, fitting into the new taxonomy of deceptive patterns. We hope that the unified taxonomy, multimodal detection approach developed in this paper as well as unexplored security implications can navigate the disruptions within the ever-evolving realm of Internet deception.

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

# APPENDIX

## A    EXISTING DP DATASETS

To raise awareness about deceptive patterns, several instances have been collected and posted on websites [7, 16, 22, 41], social platforms [6], and YouTube videos [43]. To the best of our knowledge, Mathur *et al.* [27] is the first study to systematically construct a large-scale deceptive pattern dataset. In this work, 11,000 shopping websites were analyzed, resulting in the collection of 1,818 deceptive pattern instances in the AGM-D dataset. Geronimo *et al.* [11] analyzed 240 popular mobile apps by simulating end-user interactions with these apps, recording them for analysis, and released the deceptive pattern instances as the LLE dataset. AidUI constructed context-DP [26] by collecting 175 mobile images from LLE and 83 website images in the wild. Additionally, UIGuard [8] constructed a dataset with 4,999 manually labelled non-deceptive images and 1,353 deceptive images of 1,660 deceptive instances from 1,023 mobile app. However, existing deceptive pattern (DP) datasets cannot achieve both comprehensiveness (*e.g.,* UIGuard and LLE focus on single platforms) and large scale (*e.g.,* less than 500 examples in AidUI) simultaneously. Furthermore, due to the aforementioned concept drift issue, the examples collected in previous research may become outdated and may not represent new deceptive pattern practices. Therefore, in this study we constructed a new dataset by merging and relabeling existing datasets and incorporating new DP examples from additional resources.

## B    NEW DATASET CREATION

We constructed a new DP dataset (SADP) by (*i*) merging and annotating existing datasets and (*ii*) incorporating new, up-to-date DP examples from additional resources. The statistics of the newly created dataset are reported in Table 3.

**Merging existing datasets.** We gathered 5,059 UI images including 4,906 mobile app screenshots and 153 website interfaces from 3 existing DP datasets, *i.e.,* UIGuard [8], AidUI [26], and LLE [11]. Specifically, in UIGuard, we manually checked and removed duplicate and high-similarity UI images, and corrected some labels. Finally, we selected 1,204 out of 1,353 deceptive pattern images, including 2,253 deceptive pattern instances, and 2,757 out of 4,999 non-deceptive pattern images. We included all of the 471 UI images from AidUI and re-labeled them, resulting in 499 DP instances from 302 images and 169 non-DP instances from 169 images. We obtained 173 videos from LLE and extracted DP images according to the timestamps of the deceptive patterns. To address the image overlap between LLE and AidUI, we first merged all the LLE examples into our dataset and manually eliminated duplicate and similar images through a review process Finally, we collected 627 images, of which 150 are non-deceptive images and 477 are deceptive images with 865 deceptive pattern instances.

**Incorporating new DP examples.** All the existing available datasets primarily focus on mobile platforms. However, we recognize that deceptive patterns can also exist on website platforms. There is one related work that performs deceptive pattern collection on shopping websites [27], but most of the products have been removed, and this dataset focuses only on the shopping category.

For these reason, we incorporated additional measures, such as collecting UI images from WebUI [44] (a dataset containing 400,000 webpage screenshots) and from popular lists (*i.e.,* Google Play [14] for mobile apps and SimilarWeb [38] for websites.), to enhance the number of website images and ensure that our dataset is up-to-date. We obtained 1,666 images from this process. Specifically, we manually checked 2,000 randomly sampled images from WebUI, resulting in 942 images, of which 1,806 contained deceptive patterns. Furthermore, we manually collected and analyzed 50 apps and 50 websites from Hong Kong, Australia, Ireland, and the United States in May 2024. We focused primarily on the moments when deceptive patterns appeared. After images collection and manual review, as a result, we collected 724 images, of which 722 include deceptive patterns and 2 are non-deceptive.

**Dataset annotation.** Our annotation team consists of 3 annotators and 1 advisor who is an expert in deceptive patterns. After training on the knowledge and definitions of DP, the advisor evaluated the annotators' performance on 50 randomly selected samples from all DP categories. The average annotation accuracy was 88%. The team then convened to discuss and resolve any annotation conflicts, refining their labeling standards. With the refined standards in place, each annotator independently annotated the entire dataset. At the end of the annotation process, the team collaboratively examined conflict cases and voted to determine the final label.

**Dataset Statistics Summary.** As shown in Table 2, our dataset DPGuard includes 6,725 UI images and 10,421 deceptive pattern instances, addressing the issue of scale. To ensure comprehensiveness, our dataset features 5,269 images from mobile platforms, containing 7,184 instances, and 1,456 images from websites, containing 3,237 deceptive pattern instances. The dataset spans images from 2017 to 2024, ensure it is up-to-date. Additionally, our dataset also ensure every category has at least 5 representative examples, which reduces the learning curve for future research.

## C    DATASET FOR DPGUARD EVALUATION

In DPGuard, there are two components: the binary classifier and the mutation-based prompt fine-tuned MLLM. To evaluate the performance of these two components, we used the dataset we constructed in Section B. Additionally, we removed all examples in the `Sneak into Basket` and `Tricked Question` category because examples for these two labels are rare, and the related work on these categories is too limited. Finally, we collected 3,348 deceptive UI images and 3,377 non-deceptive UI images.

**Binary Classifier Dataset**: To reduce the impact of class imbalance bias on our binary classifier, we used full deceptive pattern dataset, in which the ratio of non-deceptive to deceptive examples is nearly 1:1, to fine-tune a pre-trained convolution neural network. For fine-tuning, we split the dataset into training, validation and testing sets with a ratio of 6:2:2, using random seed 42. We ran the training for 10 epoch.

**Prompt Mutation Dataset**: For the prompt mutation part, we considered the effectiveness of our binary classifier and the inference cost. We reduced the dataset for non-deceptive UI images by taking the average number of all the deceptive pattern images across 21 classes as our target number for non-deceptive UI images. After filtering, we reserved 20% of the dataset as our testing set and used the remaining 80% for fine-tuning in each round. For each round,

we followed the setup of Promptbreeder [13], which randomly samples a batch of 100 examples as batch examples. To better fit our task, we applied a special strategy: we selected at least 5 examples from each category to obtain a batch of 100 UI Images. This strategy ensures each category is represented in the batch.

## D  EMPIRICAL EVALUATION IN THE WILD

### D.1  Methodology

For the empirical study, we first collected Android application packages (APKs) for mobile and domains for websites. Then, we employed UI exploration tools to obtain screenshots from the collected APKs and domains. To ensure the quality of the collected UI images, we designed two different strategies for mobile and website data post-processing.

For mobile images, we designed a three-step process to eliminate undesired screenshots. In Step 1, we eliminated APKs that were not able to run or collect screenshots. In Step 2, we removed highly similar images within the same app by setting a similarity threshold. In Step 3, we removed common images that did not contain any useful information (*e.g.,* blank screens, app info pages, *etc.*) by another similarity threshold.

For website images, before performing UI exploration, we sent HTTP GET requests to verify that each website returned a 200 OK status code. After that, we performed UI exploration and captured a screenshot for the visited URL.

During the screenshot collection stage, we used the code provided by WebUI [44] as a template, where WebUI employs crawlers to capture website screenshots from a list of domains. We made four modifications to adapt it to our specific goals. First, we limited the exploration waiting list to websites from the same domain where the data was collected. Second, we introduced a hashmap to ensure that each domain could explore a maximum of 20 pages. Third, instead of adding all candidate webpages one by one, we shuffled and randomly selected up to five webpages to add to the list. Finally, we removed WebUI's codebase pruning strategy, allowing more webpages to become candidates for exploration.

We then used file size to filter out images that did not contain any useful information as smaller files typically contain less information. After collecting the image data from the wild, we ran DPGuard on it and randomly sampled some data for manual review to assess the actual performance of our model in the wild.

### D.2  Dataset

To provide valuable insights into deceptive pattern in the wild, we conducted an empirical study by collecting a UI images dataset consisting of 12,301 UI images, including 2,905 UI Images from 770 out of 1,000 mobile applications and 9,396 website images from 765 out of 1,000 domains.

For mobile applications, we randomly selected 1,000 Android Package Kit (APK) released in 2024, collected by AndroZoo [1], one of the largest and fastest-growing datasets of Android applications. We applied random selection because Androzoo lacks ranking information. After collecting the APKs, we initially attempted to use SceneDroid [47], the latest automated Android GUI collection tool, to gather GUI data from these 1,000 Apks. However, in our tiny experiment, we found that only 6/40 Apks successfully recorded

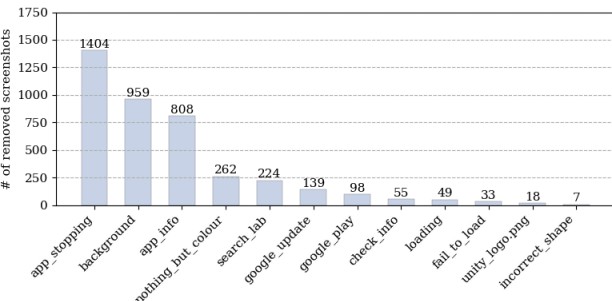

**Figure 5: Reasons of removing screenshots in step3**

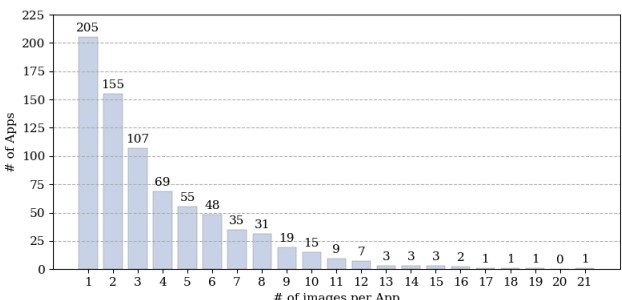

**Figure 6: Number of images distributed in an app**

screenshot. After contacting the authors, we learned that only few apps were able to collect screenshots due to the significant changes in android attack and defense environment caused by the upgrade of android [40]. As a result, we switched to Droidbot [24], for UI exploration and recording UI screenshots.

After running the 1,000 APKs, we initially collected 12,740 UI images and performed the data post-processing mentioned in Section D.1. In step1, we eliminate 41 APKs that could not run because DroidBot requires Android 9 API 28, while the eliminated APKs require at least API 29. Before proceeding to step 2 and step 3, we randomly selected 20 APKs from 7 application category and manually created a ground truth of how many images should remain in each steps which can help determine the similarity threshold for step 2 and step 3.

In step2, based on the ground truth, we expected to retain 172 images that are unique from these 20 APKs. We experimented with threshold of 0.85,0.90,0.95, resulting in 87,126 and 175 images remaining, respectively. This indicate that the best threshold for step2 was 0.95. After applying this threshold, we delete 5,779 and keep 6,961 images from step 2.

In step3, we manually reviewed 50 APKs to identify UI images with meaningless information. We conducted experiments with the same three different threshold (0.85, 0.90, 0.95) within the same 20 APKs. The ground truth in step 3 was 116 images. After setting the step 2 threshold to 0.95, we experimented with the three thresholds and found 58,103,137 images remaining, respectively. Therefore, the best threshold for stage3 was 0.90. After applying with this threshold, we eliminated 4,056 UI images and keep 2,905. The reason for elimination in step 3 is illustrated in Figure 5. In total, after running 770 app, we collected 2,905 UI Images from mobile application, and the distribution is shown in Figure 6.

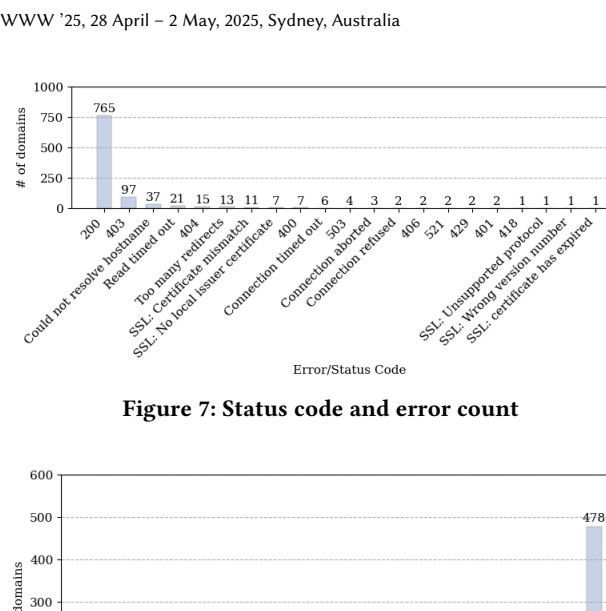

**Figure 7: Status code and error count**

**Figure 8: The number of visited webpages within a domain.**

For website images, we selected the top 1000 websites listed on Majestic Million [20], one of the most recognized website ranking list [45]. Then we executed the instruction mentioned in Section D.1. Figure 7 shows the distribution of returned status information.

After that, we followed the instructions to set a filter to only keep website images with a file size greater than 8KB because our investigation found that images smaller than 8KB are mostly incorrect responses, such as blank web pages or pages with no data. As a result, we collected 9,396 website images from 765 domains. Figure 8 provides the details on how many website images were collected per domain.

## E  INITIAL PROMPT

Now I provided you one image or sequence of images, please help me to detect whether the given image include any deceptive pattern or not.

## F  FINAL PROMPT

Kindly conduct a comprehensive analysis of the given image to identify any deceptive patterns, employing the provided taxonomy. While examining the image, consider the following categories of deceptive patterns:

1. **Nagging**: Spot any repetitive and unexpected pop-up windows that disrupt user activities. 2. **Roach Motel**: Identify scenarios where opting in is easy, but opting out is complicated or obscure. 3. **Price Comparison Prevention**: Detect any elements that hinder direct comparisons of prices or plans. 4. **Intermediate Currency**: Look for instances where virtual currencies are used to obscure real financial costs. 5. **Forced Continuity**: Examine if users are charged after a trial period ends without clear advance notice. 6. **Hidden Costs**: Check for the late disclosure of additional

costs such as taxes, delivery, or service fees. 7. **Sneak into Basket**: Identify if items not actively selected by users are automatically added to the shopping cart. 8. **Hidden Information**: Ensure that essential options or actions are not concealed or difficult to find. 9. **Preselection**: Look for cases where options are preselected by default, without explicit user consent. 10. **Toying with Emotion**: Evaluate design elements such as language, colors, or styles that are intended to elicit emotional responses and pressure users into making decisions. 11. **False Hierarchy**: Observe if any option is made to appear more significant than other equivalent choices. 12. **Disguised Ads**: Detect if advertisements are designed to resemble normal content. 13. **Tricked Questions**: Identify any confusing or misleadingly worded questions. 14. **Small Close Button**: Check if close buttons are too small to be easily located or clicked. 15. **Social Pyramid**: Look for incentives encouraging users to share content with friends for rewards. 16. **Privacy Zuckering**: Assess if default options necessitate sharing unnecessary personal information. 17. **Gamification**: Note if users are required to repeatedly perform tasks to earn rewards. 18. **Countdown on Ads**: Identify if timers restrict users from closing ads immediately. 19. **Watch Ads to Unlock**: Check if users must watch advertisements to access particular features or rewards. 20. **Pay to Avoid Ads**: Note if users are charged to remove adverts. 21. **Forced Enrollment**: Determine if users must sign up or sign in before they can utilize the service.

Additional steps for a thorough review: - **UI Color and Text**: Assess the user interface design elements such as color schemes and text that might manipulate user decisions. Focus particularly on color contrasts and overall readability. - **Information Visibility**: Verify that critical information is easily accessible without reliance on hidden links or convoluted navigation paths. - **User Experience Flow**: Evaluate the user journey to ensure smooth navigation with minimal interruptions. - **Call-to-Action Design**: Examine the visibility and clarity of call-to-action buttons and links compared to dismissive or less prominent options. - **Consistency**: Ensure that design elements and user expectations remain consistent across different pages or sections.

Your detailed review and precise identification of any deceptive patterns in this image are crucial to understand their impact and effectiveness.

## G  MUTATION CRITERIA THRESHOLD

Here we present examples of generated prompts that are relevant (irelevant) to the initial prompt in Table 7 (Table 8). For these examples, we also report the cosine similarity results to show case how we determine the mutation criteria threshold.

## H  EXPERIMENT SETTINGS

Fine-tuning the pre-trained binary classifier required the most computational resources. For this part, we utilized a machine with an Intel(R) Xeon(R) Gold 5318Y CPU @ 2.10GHz, 2 * A100 80GB PCIe GPUs, 256GB memory, and a 2TB hard drive. The prompt fine-tuning and detection were performed on a MacBook Pro M1 Max with OpenAI/Google API access. For the empirical study, we used 20TB of cloud storage to store the generated data, and the task of extracting UI images for both mobile and web platforms was also

### Table 7: Relevant Prompt Similarity Result

| Prompts | Content | Similarity |
|---|---|---|
| Initial Prompt | Here is the image, detect if any deceptive pattern in it or not. | 1.0000 |
| Prompt 1 | Assess the image for any deceptive patterns in the interface design. Confirm the absence of disruptive pop-up windows that interrupt user tasks. Ensure that all options are neutral by default, with consent checkboxes in place to avoid any unintended agreements. Look for any hidden costs that are only revealed at the final payment page. Advertisements should be clearly labeled, separate from main content, and come with accessible close buttons. Verify that all forms are clear, avoiding any confusing language. Transparency in data collection must be ensured, giving users control over their personal data, with terms and conditions easily accessible. Check the ease with which users can log out, unsubscribe, or delete their accounts. Look out for any emotional manipulation tactics such as countdown timers, fake scarcity, or confirm shaming. Evaluate overall design clarity, ensuring color schemes, text readability, and layout are user-friendly. Be vigilant for social pressure tactics or mandatory sign-ins. Confirm all relevant actions and options are visible and accessible. Ensure pricing and plan comparisons are straightforward and easily comparable with other markets. Check for transparency in the use of intermediate currencies, ensuring users understand the true monetary value. Prevent the automatic addition of items to the shopping basket. Identify any ads disguised as regular content and ensure they have large, easily located close buttons. | 0.2025 |
| Prompt 2 | Review the user interface meticulously to identify any deceptive design tactics. Ensure that all crucial information is prominently displayed in clear, legible fonts and that important links are easily accessible. Verify the absence of preselected options and that the shopping cart includes only items users have explicitly chosen. Be vigilant for manipulative strategies like countdown timers, exaggerated scarcity, and guilt-inducing language. Design buttons to be large, easy to click, and highly visible, using contrasting colors. Use clear, straightforward language, avoiding confusing constructs like double negatives. Disclose all costs, including taxes and fees, upfront and clearly. Confirm that necessary services can be accessed without mandatory sign-ups or unnecessary data collection. Document any deceptive patterns through screenshots for further analysis. Assess any gamification elements to ensure they don't enforce exploitative repetitive tasks. Check that privacy settings do not default to automatic data sharing. Ensure close buttons are sufficiently large and easy to locate. Clearly differentiate ads from regular content and avoid intrusive pop-ups. Ensure transparency with virtual currencies, making users aware of their real monetary values. Simplify the comparison process for products and plans, optimizing UI text size and color for readability and accessibility. Validate that all buttons and links are intuitive and fully operational. | 0.2516 |
| Prompt 3 | Evaluate the image for any deceptive patterns by searching for repeated pop-up interruptions, automatically selected choices that might trick users, delayed disclosures of additional costs, and advertisements that blend in as regular content. Verify the visibility and accessibility of close buttons, ensure the transparency and simplicity of any questions posed, and confirm that requests for personal data are justified. Also, review whether the color schemes, font styles, and layout arrangements serve to deceive or pressure users into unintended actions. | 0.5784 |

### Table 8: Irelevant Prompt Similarity Result

| Prompts | Content | Similarity |
|---|---|---|
| Initial Prompt | Here is the image, detect if any deceptive pattern in it or not. | 1.0000 |
| Prompt 1 | Paraphrase the original prompt and add actions loss | 0.0594 |
| Prompt 2 | Evaluate the user interface for its functionality and design elements, including layout, colors, and text to ensure user-friendliness and clarity. | 0.0827 |
| Prompt 3 | Perform an exhaustive evaluation of the user interface, with particular attention to color harmony, legibility of text, and a consistent design language. Ensure intuitive navigation with prominently positioned and clearly labeled essential buttons and features. Be vigilant for deceptive patterns including incessant pop-ups, hidden functionalities, falsely emphasized buttons, tiny close buttons, and mandatory data submission. Suggest concrete improvements to elevate user satisfaction, refine the overall user experience, and streamline the interface design. Check the efficacy of calls-to-action, confirm logical user pathways, ensure all interactions are user-friendly, and validate adherence to accessibility standards. | 0.100 |

carried out on the same MacBook Pro. For Droidbot, we set up 5 Android virtual emulators running Google Pixel 3a, featuring 4 cores, 2GB of memory, and Android 9 (API Level 29) on arm64-v8a architecture. For web UI images, we used one coordinator process and 24 threads to send the requests.

## I  EVALUATION METRICS

The deceptive pattern detection task is essentially a multi-label classification task. Like many classification tasks, we use precision, recall and F1-Score as our evaluation metric.

- Precision: The precision formula is $\frac{TP}{TP+FP}$. In our task, True Positive (TP) refers to an image that contains the specified deceptive pattern, and the model correctly predicts that the image includes the specified deceptive pattern. False Positive (FP) refers to an image that does not contain the specified deceptive pattern, but the model incorrectly predicts that it does. Precision in our task indicates how accurate the model's positive predictions are.
- Recall: The formula of recall is $\frac{TP}{TP+FN}$ where TP is the same as defined for precision, and FN refers to an image that contains the specified deceptive pattern, but the model incorrectly predicts that the image does not include the specified pattern. In our task, recall represents how well our model can identify all the positive examples.

- F1-score: The formula is $\frac{2*P*R}{P+R}$, the F1 score is used to balance the trade-off between precision and recall. It reports the harmonic mean of precision and recall, providing a single metric that reflects the balance between the two.

For each metric, we report both micro and macro average scores to demonstrate the effectiveness of our model. The micro approach calculates the total true positives, false negatives, and false positives across all classes, then computes the F1 score. This method treats every individual classification equally, regardless of the class. On the other hand, the macro approach calculates the arithmetic mean of each term for every class. It treats all classes equally by averaging the terms without considering the proportion of instances in each class.

## J  HYPER-PARAMETER

The first was to determine the threshold for mutation checker. We applied the methodology described in Section 16, and Table 9 reports the similarity between relevant/irrelevant prompt and initial prompt. Based on the result, we select 0.2 as the threshold for the mutation checker. A detailed list of the prompts and its similarity data in Appendix G. The second is to check whether the queue size affects the lifetime of each mutated prompt. Within a limited budget, we tested two queue size, and Table 10 report the

**Table 9: The similarity between the relevant/irrelevant prompt and the initial prompt. P1,P2,P3 refers to the first,second,three prompt we sampled**

|  | P1 | P2 | P3 |
|---|---|---|---|
| **Relevant** | 0.2025 | 0.2516 | 0.5784 |
| **Irrelevant** | 0.0594 | 0.0527 | 0.1000 |

**Table 10: Mutated Prompt Lifetime with Different Queue Size**

| Queue Size | Mean | Std |
|---|---|---|
| 5 | 1.6129 | 2.1253 |
| 15 | **4.4118** | **4.8332** |

queue size of 15 resulted a longer lifetime than queue size of 5.

Therefore, we selected the queue size as 15. The third experiment is about the number of rounds, within our limited budget, we set the maximum number of round to 25. The loss is defined as "1 - F1 score", "avg" refers to the average loss of all prompts in the queue, and "best" refers to the loss of the best prompt in each round. Figure 2 illustrates that our prompt mutation strategy stabilizes and reaches the minimum loss at round 24.

## K    DETAILED SOTA PERFORMANCE COMPARISON

### K.1    Mobile

### K.2    Website

## L    EXAMPLE OF DECEPTIVE PATTERNS

**Table 11: Performance Comparison between All SOTA on Mobile Platform**

| Categories | # of Instances | UIGuard | | | AidUI | | | DPGuard | | |
|---|---|---|---|---|---|---|---|---|---|---|
| | | Precision | Recall | F1 | Precision | Recall | F1 | Precision | Recall | F1 |
| No DP | 3,018 | 0.8128 | 0.8055 | 0.8091 | 0.7151 | 0.8608 | 0.7812 | 0.9830 | 0.9785 | **0.9807** |
| Nagging | 409 | 0.6462 | 0.3350 | **0.4412** | 0.2822 | 0.4450 | 0.3454 | 0.3650 | 0.4132 | 0.3876 |
| Roach Motel | 24 | - | - | - | - | - | - | 0.4474 | 0.7083 | **0.5484** |
| Price Comparison Prevention | 7 | - | - | - | - | - | - | 0.0000 | 0.0000 | 0.0000 |
| Intermediate Currency | 38 | - | - | - | - | - | - | 0.5283 | 0.7368 | **0.6154** |
| Forced Continuity | 48 | 1.0000 | 0.0208 | 0.0408 | - | - | - | 0.5915 | 0.8750 | **0.7059** |
| Hidden Costs | 38 | - | - | - | - | - | - | 0.2203 | 0.3421 | **0.2680** |
| Hidden Information | 236 | - | - | - | - | - | - | 0.4023 | 0.4364 | **0.4187** |
| Preselection | 356 | 0.3325 | 0.7187 | 0.4546 | 0.2855 | 0.4747 | 0.3565 | 0.6549 | 0.4691 | **0.5466** |
| Toying with emotion | 84 | - | - | - | 0.1136 | 0.1786 | 0.1389 | 0.1968 | 0.7262 | **0.3096** |
| False Hierarchy | 559 | 0.4324 | 0.4061 | 0.4188 | 0.7619 | 0.0286 | 0.0552 | 0.6277 | 0.6816 | **0.6535** |
| Disguised Ad | 883 | 0.5923 | 0.0872 | 0.1520 | 0.5854 | 0.1631 | 0.2551 | 0.7937 | 0.9105 | **0.8481** |
| Small Close Button | 747 | 0.9897 | 0.8969 | **0.9410** | - | - | - | 0.6785 | 0.3842 | 0.4906 |
| Social Pyramid | 35 | 0.7143 | 0.5714 | **0.6349** | - | - | - | 0.3750 | 0.7714 | 0.5047 |
| Privacy Zuckering | 206 | 0.7162 | 0.7608 | **0.7378** | - | - | - | 0.4407 | 0.3786 | 0.4073 |
| Gamification | 27 | 0.8571 | 0.2222 | 0.3529 | - | - | - | 0.4545 | 0.5556 | **0.5000** |
| Countdown on Ads | 77 | 0.5882 | 0.1299 | 0.2128 | 0.0000 | 0.0000 | 0.0000 | 0.2568 | 0.8571 | **0.3952** |
| Watch Ads to unlock features or rewards | 67 | 0.7895 | 0.2239 | **0.3488** | - | - | - | 0.0000 | 0.0000 | 0.0000 |
| Pay to avoid ads | 106 | 0.6923 | 0.7642 | **0.7265** | - | - | - | 0.7195 | 0.5566 | 0.6277 |
| Forced Enrollment | 149 | - | - | - | - | - | - | 0.2891 | 0.9060 | **0.4383** |
| micro avg | 7,114 | 0.7151 | 0.6253 | 0.6672 | 0.5923 | 0.5855 | 0.5889 | 0.7055 | 0.7598 | **0.7316** |
| macro avg | 7,114 | 0.4165 | 0.2701 | 0.2851 | 0.1247 | 0.0978 | 0.0878 | 0.4102 | 0.5312 | **0.4385** |

**Table 12: Performance Comparison between All SOTA on Website**

| Category | Instances | AidUI | | | DPGuard | | |
|---|---|---|---|---|---|---|---|
| | | Precision | Recall | F1 | Precision | Recall | F1 |
| No DP | 359 | 0.3205 | 0.6713 | 0.4338 | 0.8512 | 0.7967 | **0.8230** |
| Nagging | 180 | 0.1923 | 0.0833 | 0.1163 | 0.4103 | 0.6222 | **0.4945** |
| Roach Motel | 13 | - | - | - | 0.2703 | 0.7692 | **0.4000** |
| Price Comparison Prevention | 27 | - | - | - | 0.3333 | 0.1852 | **0.2381** |
| Intermediate Currency | 5 | - | - | - | 0.3333 | 0.6000 | **0.4286** |
| Forced Continuity | 26 | - | - | - | 0.2222 | 0.7692 | **0.3448** |
| Hidden Costs | 99 | - | - | - | 0.2025 | 0.3333 | **0.2519** |
| Hidden Information | 377 | - | - | - | 0.4363 | 0.4721 | **0.4535** |
| Preselection | 413 | 0.4006 | 0.3317 | **0.3629** | 0.3973 | 0.2107 | 0.2753 |
| Toying with emotion | 229 | 0.5652 | 0.3406 | 0.4251 | 0.5341 | 0.6507 | **0.5866** |
| False Hierarchy | 320 | 0.6667 | 0.0125 | 0.0245 | 0.3744 | 0.5219 | **0.4360** |
| Disguised Ad | 256 | 0.3814 | 0.1445 | 0.2096 | 0.7782 | 0.8359 | **0.8060** |
| Small Close Button | 160 | - | - | - | 0.4054 | 0.1875 | **0.2564** |
| Social Pyramid | 7 | - | - | - | 0.2000 | 0.8571 | **0.3243** |
| Privacy Zuckering | 367 | - | - | - | 0.6966 | 0.5068 | **0.5868** |
| Gamification | 1 | 0.0000 | 0.0000 | 0.0000 | 0.0000 | 0.0000 | 0.0000 |
| Countdown on Ads | 10 | - | - | - | 0.2759 | 0.8000 | **0.4103** |
| Watch Ads to unlock features or rewards | 0 | - | - | - | 0.0000 | 0.0000 | **0.0000** |
| Pay to avoid ads | 7 | - | - | - | 0.1429 | 0.1429 | **0.1429** |
| Forced Enrollment | 89 | - | - | - | 0.2102 | 0.8315 | **0.3356** |
| micro avg | 2945 | 0.3621 | 0.2912 | 0.3228 | 0.4691 | 0.5328 | **0.4989** |
| macro avg | 2945 | 0.1148 | 0.0720 | 0.0715 | 0.3216 | 0.4588 | **0.3452** |

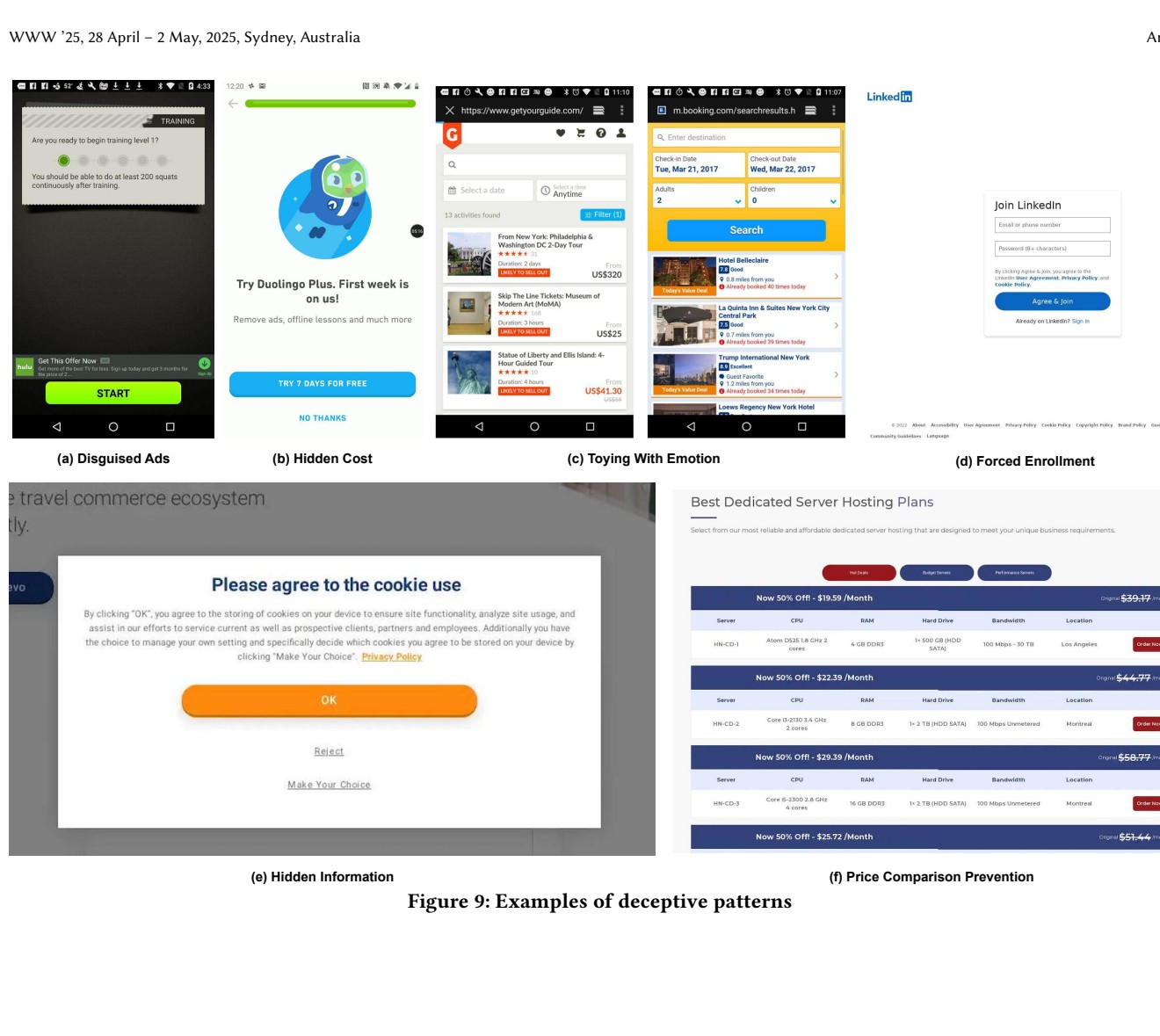

(a) Disguised Ads    (b) Hidden Cost    (c) Toying With Emotion    (d) Forced Enrollment

(e) Hidden Information    (f) Price Comparison Prevention

Figure 9: Examples of deceptive patterns

