# OpenReview forum: "50 Shades of Deceptive Patterns: A Unified Taxonomy, Multimodal Detection, and Security Implications"
_ACM.org/TheWebConf/2025/Conference — WWW 2025 Oral_

### Official Review · Reviewer_BPDY · 2024-11-14

**Novelty:** 5
**Technical Quality:** 5

**Review:**

**Paper Summary**
This paper looks at the problem of categorizing and detecting deceptive patterns (DP) in mobile apps and websites.
Specifically, the authors have compiled a comprehensive dataset of DPs, by merging several datasets from prior work and introducing new samples to the collection.
Furthermore, they have developed a new technique for DP detection, called DPGuard, which they show can outperform SOTA methods.
Lastly, they use DPGuard to detect DP instances in mobile apps and websites in the wild.

**Review Summary**
Overall, I think the paper has several interesting contributions that are novel and impactful.
However, I am not convinced that the "unified taxonomy" is as significant of a contribution as claimed by the authors and am skeptical of some of the modifications made to prior taxonomies (see detailed comments below).
On the training and evaluation of DPGuard, I also have a couple of questions, which I hope the authors can clarify.
Nevertheless, I believe that the paper holds potential and can be improved.

**Detailed Comments**
1. Unified Taxonomy. From what I can see from Table 2, the authors mostly use the taxonomy from [8] and introduced some minor changes to the definitions. While the authors have provided justifications for the modifications made, I do not see how this has "incorporat[ed] privacy and security aspects" in a way that prior work fundamentally hadn't. For instance, isn't Privacy Zuckering, which was already identified in [8], a problem precisely because of its implications for privacy? Furthermore, in some cases, I believe that the definitions of DPs in the new unified taxonomy is overly broad. For instance, I am not sure why ads in the top and bottom constitute disguised ads? They are clearly out of the way of main content and it seems like the authors are essentially saying all ads no matter where or how they appear are deceptive patterns. Perhaps it would be useful to not only provide cases with each deceptive pattern but the analogs that achieve similar functionality but would not be considered deceptive patterns. This would help to clear the confusion on where you draw the line between "acceptable functionality" and "deceptive pattern". Overall, it is my opinion that the authors have oversold their contributions to the taxonomy and the modifications have not always been well-founded.

2. Dataset. Interestingly enough, I believe that the authors have undersold their contributions in terms of compiling the dataset, relegating most of the details of this contribution to the Appendix. I think its a significant contribution that the authors have manually annotated 3,430 instances of DP, especially from websites. Therefore, I believe that the paper will greatly benefit from discussing more details about the dataset collection, compilation, analysis in the main body of the paper and toning down their claims about the taxonomy instead.

3. Experimental Evaluation. I find the details surrounding the experimental evaluation of DPGuard to be slightly confusing. For instance, I can't find details on what is the exact dataset used to train/test the prompts for the MLLM model and evaluate the overall performance of DPGuard. Or on whether the existing deceptive pattern tools (UIGuard, AdUI) were trained on the newly compiled dataset. Please see detailed questions below. Therefore, I would also appreciate a clearer explanation of the details involving the DPGuard evaluation. In my opinion, the simplest way to clear this is by first splitting the 6,725 samples (3,348 DP) into train/validate/test and only use the train & validate sets for both binary classification training and prompt engineering. Subsequently, report all results (precision/recall/f1) with respect to the common test set.

4. Case Study. Again, similar to the "unified taxonomy" i am not completely convinced that the case studies are a significant contribution. I understand that the authors would like to discuss the privacy and security implications of DP, but these case studies seem to be mostly hypothetical "Alice and Bob" scenarios. Unless the authors have specifically conducted user studies where they can discuss these implications in more concrete ways, I do not see why it is useful to consider these implications.

5. Implications. Although the authors mention briefly that detecting DP can "mitigate the risk of deception", I am not completely sure that the authors have explained _how_ this might be possible.

**Questions:**

1. Did the authors try to fine-tune the ResNet101 model on the DP categorization task? I am not sure why they used an MLLM, when the ResNet model significantly outperformed the MLLM for DP detection
2. Could the authors explain why are the MLLMs (e.g., GPT-4o) so ineffective in detecting DP but so effective in categorizing DP? Isn't categorization a harder task than detection?
3. Using commercial LLMs significantly reduces the impact of DPGuard because categorization has to be outsourced. Have the authors tried using open-source LLMs instead?
4. Why was a 1:1 ratio (DP : non-DP) used to evaluate the models' performance when in the real world we do not expect 1:1 DP to non-DP?
5. I would appreciate if the results are not only given in terms of "DP instances" but also in terms of the number of apps and websites actually containing DP. For instance, under Section 5.2 could the authors clarify out of the 1,000 mobile apps and 1,000 websites, how many of them contained DPs?
6. Could the authors publish their compiled dataset for future research?

**Reviewer Confidence:**

3: The reviewer is confident but not certain that the evaluation is correct

**Scope:**

4: The work is relevant to the Web and to the track, and is of broad interest to the community

---

### Official Review · Reviewer_YyoV · 2024-11-29

**Novelty:** 6
**Technical Quality:** 6

**Review:**

This is a clearly written and important peice that has clear relevance to current concerns within the ACM webConf community. It makes an important contribute to understanding how deceptive design patterns impact and undermine trust online, and how these might be detected. The large dataset that the paper draws on shows that deceptive design practices are widespread, and as a result the paper raises interesting broader questions about how such practices have become normalised within app and website development.

The dataset built for the paper, and the DPGuard framework, are extensive and well-developed. The hybrid detection approach that the paper describes, which combines binary classifiers with MLLMs, seems to usefully address limitations of rule-based systems and would allow the scalability of the DPGuard tool.

**Questions:**

The dataset runs from 2017 to 2024, but how will the DPGuard tool be adapted as developers implementing deceptive practices also adapt? Relatedly, how is the pattern of false positives and negatives distributed? Is the impact of these equitable on different types of developers - for example, are some likely to be better resourced to evade detection, and how can this be accounted for?

As a relatively unfamiliar reader, the framework and development process was challenging to follow - how can the author(s) ensure that the tool is transparent, interpretable and useful to end-users like regulators? Has there been engagemnt with end users, or is there a plan to do so to ensure that the tool is used?

The use of LLMs here will incur substantial computational (and energy) costs - is there a way to offset this?

**Reviewer Confidence:**

2: The reviewer is willing to defend the evaluation, but it is likely that the reviewer did not understand parts of the paper

**Scope:**

4: The work is relevant to the Web and to the track, and is of broad interest to the community

---

### Official Review · Reviewer_54rq · 2024-11-30

**Novelty:** 4
**Technical Quality:** 4

**Review:**

This paper presents a unified taxonomy of deceptive patterns with 24 refined subcategories, addressing gaps in existing frameworks. It introduces a novel detection approach combining binary classification and mutation-based prompt engineering with multimodal large language models, achieving state-of-the-art performance through DPGuard. The authors also provide real-world cases mapped to the new taxonomy.

### Strengths

- New detection approach combining binary classification and mutation-based prompt engineering with multimodal large language models
- The performance of DPGuard is improved compared to the existing works
- Real-world cases mapped to the new taxonomy

### Weaknesses

- Some deceptive patterns are exaggerated
- The source of the Android app dataset is not the app markets

### Details

I really appreciate the method design and evaluation experiments in this paper. The author introduced multimodal large language models into deceptive pattern detection and achieved state-of-the-art performance, proving the effective application of new technologies in traditional fields, which is commendable. In addition, I was also deeply impressed by the evaluation experiments in this paper, which were well-designed and reasonable. Moreover, this paper lists many real-world cases for illustration, which can help readers understand this paper well.

However, I do not agree with some of the deceptive patterns and related security risks listed by the author in the paper. For example, Countdown on Ads (Category 17) is a common advertising model on mainstream video websites (such as YouTube). I think it is an exaggeration to classify this advertising model as deceptive patterns. In addition, the author emphasizes mandatory registration (Category 21), which I think is necessary for social apps (such as LinkedIn, as cited by the author). Social apps aim to facilitate users to establish communication, so account registration is reasonable. Moreover, app service providers can also increase the difficulty of automatically collecting users' privacy information through mandatory account registration, which protects users' privacy information from large-scale leakage. For the case of Price Comparison Prevention cited by the author, I think it is a normal function designed for different customer groups and does not pose a risk to user interests. There is also a similar case of Toying With Emotion, which should also be a normal commercial marketing behavior. The author should further optimize the deceptive patterns and filter out patterns or application scenarios that will not cause real risks.

The Android app dataset in the paper comes from Androzoo, a public experimental dataset, not app markets. This situation will affect the empirical evaluation of deceptive patterns in the wild. Considering the difficulty of collecting apps in the app markets, the author can collect app information, such as popularity, classification, etc., and then obtain the corresponding APK file from Androzoo.

**Questions:**

- Are there any real-world examples that illustrate the security risks of some deceptive patterns, such as those mentioned in Details?
- Why was the app market not considered when building the Android app dataset?

**Reviewer Confidence:**

3: The reviewer is confident but not certain that the evaluation is correct

**Scope:**

3: The work is somewhat relevant to the Web and to the track, and is of narrow interest to a sub-community

---

### Official Review · Reviewer_EPz4 · 2024-12-01

**Novelty:** 4
**Technical Quality:** 5

**Review:**

I enjoyed reading your paper; you are working on a pertinent and timely topic. Deceptive patterns are everywhere, and, as you also wrote in your paper, it is hard to keep up with all the emerging deceptive techniques. The topic of the paper is relevant to this community: both the focus on web and privacy are in line with the track's interests, and the machine learning solution for detecting deceptive patterns might be of interest to the rest of this conference's audience. I appreciate the extensive data set and experiments and I like the attempt to look at DPs in the wild. However, the contribution to the deceptive patterns taxonomy is rather limited, i.e., you mostly reuse categories from prior work and the security and privacy examples you add are very weak. Additionally, the detection approach is not very novel, i.e., it mostly uses state-of-the-art building blocks, but the evolution of the prompts is an intriguing fresh idea. Also, the in-the-wild results are hard to trust because of the poor performance of the classifier and the lack of manual validation. Moreover, I would have liked to see more concrete examples in the case studies (Section 6): give examples of concrete websites employing that particular deceptive pattern, how common the pattern is in the wild, would prior taxonomies consider it accordingly, are users routinely tricked by the discussed DP, etc.

Pros
+ Large dataset with both mobile and web samples. I appreciate the manual annotation effort and curating of the existing data sets, but the data set is mostly a merge of the prior ones proposed in the literature.
+ Intriguing idea to evolve prompts,
+ I liked the manual pilot study and the approach to calculating cosine of generated prompts.

Cons
- Small delta to prior taxonomies,
- The proposed detection approach is very noisy. Results in the wild (Section 5.2) are very hard to trust because of the poor performance of the classifier,
- The security and privacy discussion is weak,
- The main body of the paper needs example prompts that show the benefit of evolving prompts. Also, the need for the LLM-part of the DP-Guard is insufficiently motivate. That is, why not using multiple binary classifiers, one per category?

Further comments on clarity
Your paper contains many incorrect statements, such as "This lack of trust introduces vulnerabilities." It also shows a peculiar understanding of the web security model. How exactly does clicking on disguised ads lead to "malware installation?" Attackers can rarely escape the browsers' sandbox these days. Additionally, the whole Alice and Bob examples are very hard to read for a security person, i.e., a lot of imprecise and incorrect statements. Concretely, I found these statements problematic:
* "For the privacy risk, Bob prevents Alice from comparing prices might force her to engage more with Bob's system, potentially collecting more of her personal data without her realizing the extent of the tracking."
* "it exposes Alice to financial exploitation and unauthorized data sharing, making her more vulnerable from both security and privacy perspectives."
I recommend adding a threat model in the beginning of the paper to show how adversaries can weaponize deceptive patterns, instead of these Alice and Bob examples. Also, please explain what you mean by "micro" and "macro" results in the main body of the paper, not only in the appendix.

**Questions:**

1. Why did you include so few non-DP websites in your dataset (See Table 3)? Doesn't this bias the binary classifier?
2. How exactly does your dataset reduce the learning curve for future research? Who is your target audience, e.g., new researchers in the field? How is the data set used by humans?
3. Can you provide more information about the prompts quality checker? How do you ensure that "prompts are of high quality"? As I understand, this step is manual, right?
4. Please explain what you mean by "GPT's randomness in prompt generation". How do you configure/trigger this randomness? What impact does this have on reproducibility

**Reviewer Confidence:**

3: The reviewer is confident but not certain that the evaluation is correct

**Scope:**

4: The work is relevant to the Web and to the track, and is of broad interest to the community

---

### Official Review · Reviewer_wxUA · 2024-12-01

**Novelty:** 5
**Technical Quality:** 3

**Review:**

The paper investigates deceptive patterns (DPs), which are user interface designs aimed at manipulating users into unintended decisions for commercial or service benefits. It addresses the challenges in identifying deceptive patterns by refining a taxonomy from security and privacy perspectives. The authors created a large-scale dataset of 6,725 images and 10,421 instances from mobile apps and websites. They developed "DPGuard," an automated detection tool utilizing multimodal large language models (MLLMs), which outperformed existing methods. Empirical evaluation on 2,000 apps and websites revealed widespread use of deceptive patterns, with 23.61% of mobile screenshots and 47.27% of website screenshots containing at least one DP instance. The study highlights the taxonomy's importance for addressing internet deception challenges.

## Strengths

1. The paper is well-structured and clearly written, making it easy to follow the authors' arguments and experimental setup.
2. Integrating multimodal large language models (MLLMs) into the investigation of deceptive patterns represents an excellent attempt to use advanced multimodal models to tackle traditional tasks.
3. The paper demonstrates substantial effort by collecting a large and effective dataset of deceptive patterns (DPs).

## Weaknesses

1. The use of ResNet as the final binary classification model raises concerns about its ability to handle data from new categories.
2. The method of incorporating domain-specific knowledge into the revised prompting strategy needs further refinement. Would it be more effective to directly let the large model select the appropriate strategy and adapt to subtasks?
3. The strong performance of networks like ResNet in binary classification tasks might result from suboptimal dataset splits, potentially leading to a "magic trick" effect. This casts doubt on its generalization ability.
4. The study does not consider more direct large language model (LLM) prompting strategies, such as selecting graphical examples for in-context learning to improve model performance.

**Questions:**

1. **What are the specific operations of the mutation module?**
    - Could you provide detailed information about the prior knowledge given to the model for this module? This is essential to confirm whether the design is reasonable and aligns with the task's objectives.
2. **Why does ResNet outperform other models?**
3. **What is the final output of the MLLM?**
    - Does the MLLM provide interpretable responses? If so, could you elaborate on how interpretability is achieved and whether it contributes to better understanding the model’s decision-making process?

**Reviewer Confidence:**

3: The reviewer is confident but not certain that the evaluation is correct

**Scope:**

3: The work is somewhat relevant to the Web and to the track, and is of narrow interest to a sub-community